# Assessment of Disordered Linker Predictions in the CAID2 Experiment

**DOI:** 10.3390/biom14030287

**Published:** 2024-02-28

**Authors:** Kui Wang, Gang Hu, Zhonghua Wu, Vladimir N. Uversky, Lukasz Kurgan

**Affiliations:** 1School of Statistics and Data Science, LPMC and KLMDASR, Nankai University, Tianjin 300071, China; wangkui@nankai.edu.cn (K.W.); huggs@nankai.edu.cn (G.H.); 2School of Mathematical Sciences and LPMC, Nankai University, Tianjin 300071, China; wuzhh@nankai.edu.cn; 3Department of Molecular Medicine, USF Health Byrd Alzheimer’s Research Institute, Morsani College of Medicine, University of South Florida, Tampa, FL 33613, USA; vuversky@usf.edu; 4Department of Computer Science, Virginia Commonwealth University, Richmond, VA 23284, USA

**Keywords:** intrinsic disorder, disordered linkers, protein structure, protein function, prediction, assessment

## Abstract

Disordered linkers (DLs) are intrinsically disordered regions that facilitate movement between adjacent functional regions/domains, contributing to many key cellular functions. The recently completed second Critical Assessments of protein Intrinsic Disorder prediction (CAID2) experiment evaluated DL predictions by considering a rather narrow scenario when predicting 40 proteins that are already known to have DLs. We expand this evaluation by using a much larger set of nearly 350 test proteins from CAID2 and by investigating three distinct scenarios: (1) prediction residues in DLs vs. in non-DL regions (typical use of DL predictors); (2) prediction of residues in DLs vs. other disordered residues (to evaluate whether predictors can differentiate residues in DLs from other types of intrinsically disordered residues); and (3) prediction of proteins harboring DLs. We find that several methods provide relatively accurate predictions of DLs in the first scenario. However, only one method, APOD, accurately identifies DLs among other types of disordered residues (scenario 2) and predicts proteins harboring DLs (scenario 3). We also find that APOD’s predictive performance is modest, motivating further research into the development of new and more accurate DL predictors. We note that these efforts will benefit from a growing amount of training data and the availability of sophisticated deep network models and emphasize that future methods should provide accurate results across the three scenarios.

## 1. Introduction

Intrinsically disordered regions (IDRs) are segments in protein sequences that lack stable tertiary structure under physiological conditions [1]. While cellular functions of many IDRs involve transitioning into a structured state(s), the entropic disordered regions function by shifting between conformational states without becoming structured [2,3]. The most commonly annotated type of entropic disordered regions in the DisProt database [4,5,6,7], which is the main source of functionally annotated IDRs [4], are the disordered flexible linkers (DLs). They are defined as regions that connect, provide separation, and permit movement between adjacent functional regions, which could be structured domains or disordered motifs (IDPontology term: 00503 [4,8]). DLs are distinct from flexible linkers, which are flexible (but not necessarily disordered) inter-domain regions (e.g., hinges) that allow domains to move relatively to each other [9] in several ways: they are intrinsically disordered, longer, and localized in both inter-domain and intra-domain fashion [10]. DLs contribute to many cellular functions, with just a few examples that include allosteric regulation [11], peptide aggregation [12], phase separation [13], and DNA packaging [14]. Furthermore, some DLs act as entropic clocks that are crucial for timing various cellular processes, e.g., the timing of the intra- and inter-molecular binding events involved in gating and clustering processes of the voltage-activated potassium channels [15] via the so-called “ball and chain” mechanism [16,17]. Long DLs contribute to the functionality of stochastic machines, which are protein complexes formed via the fusion of proteins with flexible linkers that can dramatically accelerate chemical interactions between them by their colocalization and random movements and not via coordinated conformational changes [18].

Although it may seem that any IDR can act as a linker [19], the levels and depth of disorder in proteins can be very different, with different “disorder flavors” being distinguished based on differences in amino acid compositions, sequence locations, and biological functions [20]. Even at the global level (i.e., at the level of a whole protein or protein domain), one can clearly distinguish compact (molten globule-like) and extended disorder (coil-like and pre-molten globule-like) [21,22]. Furthermore, since different parts of a protein can be (dis)ordered to a different degree, a protein molecule may exist as a dynamic structural ensemble of foldons, inducible foldons, inducible morphing foldons, nonfoldons, semifoldons, and unfoldons, with all these differently (dis)ordered elements possessing different specific functions [2,3]. In other words, the intrinsic disorder phenomenon underlies the structure–function continuum model linking structural heterogeneity and multifunctionality of proteins. Here, independently foldable units or foldons of a protein can contribute to the catalytic and transport functions, whereas IDRs can (partially) fold when interacting with binding partners (inducible foldons) or even fold differently upon interaction with different binding partners (inducible morphing foldons) or could be in a semi-folded form (semifoldons). Moreover, ordered protein regions that undergo an order-to-disorder transition to become functional (i.e., unfoldons) are related to a vast set of dormant or cryptic disorder-associated functions [23]. Finally, various entropic disordered region activities, including DLs, are ascribed to the nonfoldable protein regions or nonfoldons.

Looking at this structural and functional heterogeneity of IDRs [2,3], it is clear that specific computational methods are required if one would like to find specific disorder-based functions in query proteins. Recent bioinformatics analysis suggests that thousands of proteins are likely to have DLs, including about 7% of human proteins that were predicted to have one or more long DL regions that are at least 20 consecutive residues in length [10]. Given that only about 400 experimentally annotated DLs are currently known [4], several computational predictors that identify these regions in protein sequences were developed [24]. They include DFLpred [10], APOD [25], and TransDFL [26]. These predictors were trained to predict DLs defined in the DisProt database (IDPontology term: 00503). At the same time, well over 100 methods that predict a more generic class of disordered regions were published [24,27,28,29,30]. The predictive performance of these methods was evaluated in two large and recently completed community-driven Critical Assessments of protein Intrinsic Disorder prediction (CAID): CAID1 [31,32] and CAID2 [33]. These assessments are performed by independent assessors (who exclude authors of the evaluated methods) using large and blind benchmark datasets (authors of predictors have no access to these data before the assessment), community-accepted protocols and metrics, and predictors that are provided by the authors to the assessors before the experiment. This makes these assessments arguably more objective and reliable compared to smaller-scale evaluations that are done when individual methods are published and evaluated by the authors of these methods.

In particular, CAID2 was the first assessment that included an evaluation of the predictions of DLs. The organizers used a dataset of 40 proteins that have DLs to comparatively evaluate 41 predictors, which include two DL predictors and 39 disorder predictors [33]. The results, which are summarized in Figure 3C,D in ref. [33], suggest that several methods perform relatively well. Specifically, SPOT-Disorder2 [34], SETH [35], and Dispredict3 (unpublished) are shown to secure the Area Under the ROC Curve (AUC) of 0.782, 0.770, and 0.744, respectively. However, this analysis considers a rather narrow scenario where these methods are applied to predict residues in DLs in proteins that are already known to have DLs. Moreover, DL residues are the only disordered residues for 45% of the 40 proteins that were used in this evaluation, essentially allowing generic disorder predictors to correctly identify disordered linkers. In other words, this evaluation does not address a broader scenario where residues in DLs must be separated from the majority of other types of disordered residues. Motivated by the availability of quality benchmark data and predictions from the CAID2 experiment, we evaluate predictions of DLs by considering several arguably more practical scenarios. First, we test accuracy when making predictions of residues in DLs for a broad collection of over 300 proteins that include a variety of different types of intrinsically disordered residues, including the 40 proteins harboring DLs. Second, we specifically evaluate whether the current methods can accurately differentiate between residues in DLs and other types of disordered residues. Third, we assess whether those methods are able to distinguish proteins harboring DLs from other proteins that do not harbor DLs. This is a useful scenario for methods that struggle with predicting the correct positions of DL residues but which accurately identify the presence of these residues in a given sequence, especially if they make these predictions very quickly, facilitating applications to large collections of proteins. Following CAID2, we evaluate both the DL predictors and generic disorder predictors in the context of the DL prediction assessment. The inclusion of the latter group of methods is also motivated by the fact that DLs can be seen as IDRs on the extreme end of the intrinsic disorder spectrum, as they are nonfoldons. This means that disorder predictors might be able to predict residues in DLs with higher propensities for disorder than other types of IDRs that undergo some form of folding (e.g., inducible foldons, inducible morphing foldons, or semifoldons).

## 2. Materials and Methods

We use the full CAID2 test set available at https://caid.idpcentral.org/challenge, which we accessed on 5 October 2023. This dataset includes 348 proteins and 287,020 residues. Among the 348 test proteins, 40 (11.5%) have DLs. There are 37,072 intrinsically disordered residues (12.92% of residues are disordered), including 2,023 DL residues, i.e., 5.5% of the disordered residues are disordered linkers.

We consider all methods that participated in CAID2. However, we eliminate methods that achieve below 90% coverage, i.e., they are unable to make predictions for more than 10% of proteins. This could be because they can only be applied to protein sequences of a certain length and/or may not work for sequences that have nonstandard amino acids. Keeping these methods would lead to comparisons performed on substantially different datasets (different collections of proteins), which would adversely affect reliability of the corresponding observations. We note that the two DL predictors that participated in CAID2, DFLpred [10] and APOD [25], were able to predict the 348 test proteins (100% coverage). We cannot include the third DL predictor, TransDFL, since its authors did not deposit it to CAID2, and consequently, it cannot be evaluated using a fair setup where the benchmark dataset is equally unknown to the authors of the methods, i.e., some of the CAID2 test proteins could potentially be used to train the TransDFL model. In total, we assess a comprehensive collection of 37 methods that include the two DL predictors and 35 predictors of disorder. The latter predictors are (alphabetically) AIUPred (unpublished method by Zsuzsanna Dosztányi), AUCpreD [36], DeepIDP-2L [37], two versions of DisEMBL (DisEMBL-dis465 and DisEMBL-disHL) [38], DisoMine [39], DisoPred (unpublished method by Min Li), DISOPRED3 [40], Dispredict2 [41], Dispredict3 (unpublished method by Md Tamjidul Hoque), three versions of ESpritz (ESpritz-D, ESpritz-N, and ESpritz-X) [42], four versions of flDPnn (flDPnn, flDPtr, flDPlr2, and flDPnn2) [43], FoldUnfold [44], IDP-Fusion (unpublished method by Bin Liu), IsUnstruct [45], IUPred3 [46], Metapredict [47], MobiDB-lite [48], two flavours of PredIDR (PredIDR-long and PredIDR-short; unpublished methods by Kun-Sop Han), PreDisorder [49], pyHCA [50], rawMSA [51], RONN [52], s2D (unpublished method by Michele Vendruscolo), two versions of SETH (SETH-0 and SETH-1) [35], SPOT-Disorder [53], SPOT-Disorder-Single [54], and VSL2 [55]. Brief descriptions of these methods are available at https://caid.idpcentral.org/overview. We collected their predictions from https://caid.idpcentral.org/challenge.

The above methods predict propensities for DLs (for DFLpred and APOD) and intrinsic disorder (for the other 35 methods) for each residue in an input protein sequence. We evaluate these predictions by comparing these propensities against the ground truth annotations of DLs. We rely on the same metrics that were used in the CAID1, CAID2, and other recent assessments of disorder predictors [28,31,33,56,57,58,59,60]. They include AUC, lowAUCratio, and Area Under the Precision-Recall Curve (AUPRC), which quantify the performance of the predicted propensities. The lowAUCratio focuses on a part of the ROC curve where the number of residues predicted to be in DLs is relatively low (conservative) and does not exceed the number of native DL residues. It is calculated as the AUC for false positive rates that are lower than the rate of native DL residues, which is divided by the corresponding AUC of a random predictor. Consequently, a lowAUCratio > 1 means that the corresponding predictions outperform a random result, where a value of 2 denotes twice better AUC. Moreover, we compute the F1 and Matthews Correlation Coefficient (MCC) that evaluate the performance of binary predictions (i.e., linker vs. non-linker). Following CAID1 [31], we compute F1max and MCCmax. These are the maximal values of F1 and MCC established by binarizing the predicted propensities with different thresholds and selecting the threshold that produces the highest value of F1 or MCC. Larger values for each of these four metrics indicate that the accuracy of the underlying predictions is higher. We perform sampling to facilitate direct comparisons of the measured metrics between experiments. Specifically, we keep all residues in DLs and randomly sample the residues in non-DLs to obtain the content/fraction of the DL residues that is equal to the content/fraction of intrinsically disordered residues in the CAID2 dataset. This way, we can compare the quality of the linker predictions in the various scenarios that we consider (prediction residues in DLs vs. in non-DL regions; prediction of residues in DLs vs. other disordered residues) with the quality of the disorder predictions from the CAID2 experiment. Finally, we perform statistical significance tests between the leading/most accurate method and the other considered methods. This test assesses whether the improvements offered by the best method are robust across diverse datasets. To do that, we sample 50% of test proteins 10 times, and we compare the corresponding 10 results for a given pair of methods. We use the Anderson–Darling test at 0.05 significance to test whether the resulting measurements are normal; for normal measurements, we assess significance using the *t*-test; otherwise, we utilize the Wilcoxon test. While such tests are commonly used in disorder prediction assessments [58,59,60], they were not reported in CAID2, where only the overall/dataset-level scores are available.

## 3. Results

### 3.1. Prediction of Residues in Disordered Linkers in Protein Sequences

We assess the quality of predictions of residues in DLs for the 37 methods on the test dataset that includes a variety of proteins harboring DLs and other types of disordered regions. We find that the predictive quality ranges between modest (AUC between 0.70 and 0.72, and AUPRC between 0.25 and 0.3) and poor (AUC < 0.55 and AUPRC < 0.15), see Table 1. The best-performing methods, which secure AUC ≥ 0.7 and AUPRC ≥ 0.25, include APOD [25], which was specifically designed to predict residues in DLs, and SETH [35]. The key characteristic of the disorder predictor SETH is that it was trained on disorder annotations from the NMR and chemical shift experiments that produce real-valued propensity for disorder, unlike the large majority of the other disorder predictors that rely on the binary (disordered vs. structured) annotations of disorder for training. The other methods have significantly and consistently lower predictive performance across the five metrics compared with the overall most accurate APOD (*p*-value < 0.05). The low predictive performance of many of these methods is because they were designed to predict disordered residues, and apparently, their predictions cannot be used to accurately predict DL residues from among other disordered and structured residues. The modest nature of the predictive quality is also clear from the moderate values of the correlation coefficients (MCCs) that are between 0.25 and 0.3 for the best-performing predictors (APOD, SETH, and PredIDR). To compare, CAID2 results (Figure 2A,B in ref. [33]) reveal that several disorder predictors, such as flDPnn2, Dispredict3, DisoPred, and flDPlr2, secure AUC > 0.8 and AUPRC > 0.5 when applied to predict intrinsically disordered residues.

The ROC and PR curves for the top ten methods from Table 1 are in Figure 1A,B, respectively. The lowAUCratio metric focuses on the arguably most practical left side of the ROC curves, which corresponds to the predictions with low false positive rates, i.e., where the number of predicted residues in DLs does not exceed the number of native residues in DLs. We note that several methods, including APOD, SETH, PredIDR, PreDisorder, and flDPlr2, secure a lowAUCratio of about 2.5 or higher. This means that they outperform a random predictor by a factor of at least 2.5 for the low false positive rate part of their ROC curves (Figure 1A).

### 3.2. Prediction of Residues in Disordered Linkers among Disordered Residues

DLs are one of the many types of IDRs, which also include regions that interact with proteins and peptides, such as MoRFs [61], regions that interact with nucleic acids and lipids, molecular recognition display sites that host PTM sites, and self-assembly regions [4]. Given the variety of disorder functions, it is important to accurately differentiate residues in DLs from other categories of disordered amino acids. We test this by evaluating the ability of the 37 methods to predict residues in DLs among the native disordered residues in the CAID2 test dataset. Results in Table 2 show that only two methods, APOD and DFLpred, are capable of identifying residues in DLs in this scenario. These two methods were designed to predict DL residues, while the other 35 methods predict all disordered residues and, thus, expectedly, cannot accurately predict residues in DLs. APOD secures an AUC of 0.72, a lowAUCratio of 3.0, an AUPRC of 0.27, and an MCC of 0.26, matching its ability to predict residues in DLs among all residues from Table 1. The disorder predictors produce near-random levels of predictive quality and are significantly less accurate than APOD across the five metrics (*p*-value < 0.05). While DFLpred also provides statistically worse results compared to APOD (*p*-value < 0.05), it is more accurate than the disorder predictors (AUC of 0.61 vs. AUC < 0.55). The corresponding ROC and PR curves are in Figure 1C,D, respectively. They reveal a substantial gap between the curves of APOD and the other 36 predictors. Altogether, this analysis suggests that currently, only APOD is capable of relatively accurately differentiating residues in DLs from other intrinsically disordered residues.

### 3.3. Prediction of Proteins Harboring Disordered Linkers

Based on the current data in version 9.4 of the DisProt database [4], about 10.6% of the intrinsically disordered proteins harbor DLs (i.e., 282 out of 2649 proteins in DisProt). This is similar to the rate of DLs in the CAID2 test dataset that we use (11.5%; 40 out of 348 test proteins). The first step in finding DL residues is to identify proteins that have these regions. We are the first to design an approach for that purpose and evaluate the current residue-level methods in this context.

We use the residue-level predictions generated by each of the 37 predictors to generate protein-level scores that can be used to identify proteins harboring DLs. DLs range between 10 and 288 consecutive residues in length, with a median length of 31, in the test dataset. Correspondingly, we process the residue-level scores to identify segments of consecutive residues with high propensities. More precisely, we apply sliding windows of size 10 (minimal DL length) and 31 (median DL length) to scan the input sequences. We compute an average of the residue-level propensities in each window and select the highest of these averages as the protein-level score. This simulates identifying a DL region in the sequence, which by definition should be composed of consecutive residues that obtain high predicted propensities. Moreover, as an alternative, we compute an average of the predicted residue-level propensities for the 10 and 20 residues with the highest propensities. These two protein-level scores focus on the subset of residues with the highest scores, as they would be a proxy for the presence of DLs. We do not consider using the average for all residues in the sequence since DLs cover a relatively small fraction of the amino acids in the sequences. We empirically compare the four protein-level scores that we define above by applying them to the predictions of the 37 methods. We pick the most accurate of the 37 methods for each protein-level score, as measured using the AUC and AUPRC of the corresponding predictions. Results, which we summarize in Figure 2, reveal that APOD generates the best protein-level scores. This agrees with the relatively best predictive quality of this method in Table 1 and Table 2. Among the four protein-level scores, the approach based on the median size sliding window produces the best performance, with AUC of 0.664 and AUPRC of 0.226. This can be explained using the fact that this approach most closely mimics the underlying characteristics of DLs, including their size and segment-based nature.

We compare the results obtained by the 37 predictors using the best-performing protein-level scores in Table 3. APOD’s predictions are the most accurate, reaching a lowAUCratio of 2.6 and an MCC of 0.22. This corresponds to low levels of correlation with the ground truth and the 2.6-fold improvement over a random predictor, respectively. Most of the other methods have significantly lower predictive performance when compared with APOD (*p*-value < 0.05), with the exceptions of DFLpred, which obtains lower AUC, PreDisorder, and s2D that secure lower lowAUCratio and AUPRC, and PredIDR that has a lower F1 and MCC, but where these differences are not statistically significant (*p*-value ≥ 0.05). The corresponding ROC and PR curves are in Figure 1E,F, respectively. These curves have a serrated shape because this evaluation is conducted at the protein level, where there are 40 proteins harboring DLs (i.e., 40 positives). While the ROC curves, in general, are better than random (i.e., above the diagonal line), only the curves for APOD and PreDisorder perform relatively well for low false positive rates where the predictions are practical (i.e., where a relatively limited number of proteins without DLs are incorrectly predicted to have them). This is why these two methods secure the highest values of lowAUCratios at 2.6 and 2.4, respectively. Overall, the analysis of Table 3 and the two curves suggests that APOD outperforms the other methods for the prediction of proteins harboring DLs; however, its predictive quality is rather modest.

## 4. Summary and Conclusions

CAID experiments provide invaluable insights into the performance of the current predictors of disorder and disorder functions [31,33]. They were recently used to identify accurate and practical methods for the prediction of disorder and disordered binding regions [62], analyze the performance of predictors that apply deep learning [58], investigate the use of AlphaFold for the disorder prediction [60,63,64], and design new methods for protein structure and disorder function predictions [47,65,66,67,68,69,70,71,72,73,74]. CAID2 featured the first evaluation of predictions of DLs [33]. However, this evaluation reflects a rather restricted scenario where methods are applied to proteins that are already known to have DLs and where DLs are the only disordered regions for nearly half of these test proteins. We substantially expand this assessment by using a much larger collection of nearly 350 test proteins from CAID2 and evaluating three distinct scenarios: (1) prediction of residues in DLs vs. in non-DLs (arguably the typical use of DL predictors); (2) prediction of residues in DLs vs. other disordered residues (to evaluate whether predictors can separate residues in DLs from other types of disordered residues); and (3) prediction of proteins harboring DLs. We summarize the results from these three scenarios in Figure 3. The x-axis shows that the DL predictor APOD and several disorder predictors, such as SETH, PredIDR, and Dispredict3, provide relatively accurate predictions of residues in DLs. However, only APOD accurately identifies residues in DLs among other types of disordered residues (y-axis) and predicts proteins harboring DLs (size of the marker). Altogether, we find that the one universally accurate method is APOD, while the disorder predictors and DFLpred perform rather poorly in identifying residues in DLs among other types of disordered residues. These conclusions are distinct from the observations in CAID2, where disorder predictors were found to be more accurate than the DL predictors when predicting residues in DLs for proteins which are known to have disordered linkers [33].

We also note that APOD’s predictive performance is moderate, with AUCs ranging between 0.66 and 0.72 and AUPRC values between 0.23 and 0.29. This observation, along with the functional importance and abundance of DLs discussed in the introduction, motivate further research into designing and deploying new and more accurate methods. Importantly, these efforts should ensure that future methods provide accurate results for each of the three scenarios. Moreover, this need is particularly urgent for identifying proteins harboring DLs, where the current predictors are limited to AUCs < 0.67 and AUPRCs < 0.23. One of the reasons for this rather poor performance is that current methods produce residue-level predictions while the latter scenario considers protein-level predictions. Designing the protein-level methods would likely lead to more accurate results, given that such predictors would benefit from a design where information from the entire protein chain is used for the prediction, in contrast to the residue-level predictors that typically rely on a local sequence window. The new methods would also benefit from larger training datasets, as the size of the DisProt database, which is the main source for the experimentally annotated DLs, continually grows from 800 proteins in version 7.0 (September 2016) to 2039 proteins in version 9.0 (September 2021) and to 2649 in the newest version 9.4 (June 2023). Accurate predictors of proteins harboring DLs would be particularly useful if their results could be produced very quickly, allowing for fast screening of proteins for the presence of DLs. Such fast predictions could be followed up by running a relatively slow APOD, which takes about 560 s to predict a 1000-residue-long sequence [33] and which accurately identifies residues in DLs. Finally, we anticipate that the recently popular deep learning models should play an important role in the efforts to design this new generation of accurate DL predictors, given the success of these models in the context of intrinsic disorder prediction [58]. The two DL predictors evaluated in CAID2 rely on other types of predictive models, including logistic regression in DFLpred [10] and support vector machine in APOD [25].

## Figures and Tables

**Figure 1 biomolecules-14-00287-f001:**
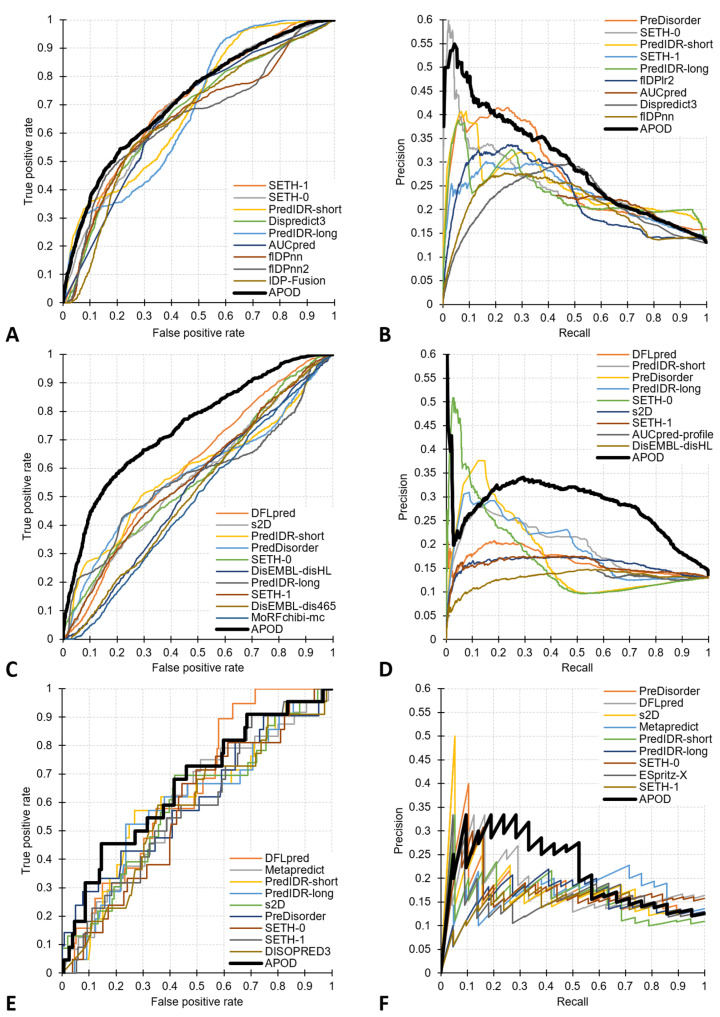
ROC curves and PR curves for the tests on the CAID2 test dataset with 348 proteins. Panels (**A**,**B**) focus on the assessment of the predictions of DL residues in sequences. Panels (**C**,**D**) are for the predictions of residues in DLs vs. other disordered residues. Panels (**E**,**F**) are for the prediction of the proteins harboring DLs. Each panel includes results for the top 10 methods for a given metric and dataset.

**Figure 2 biomolecules-14-00287-f002:**
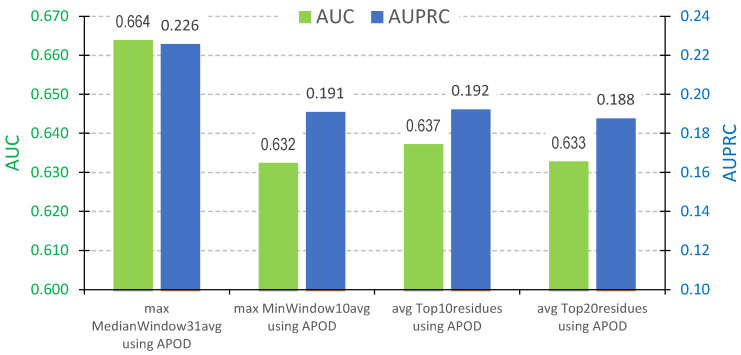
Predictive performance for the prediction of proteins harboring DLs on the CAID2 test dataset with 348 proteins. We compute the protein-level scores from the residue-level predictions using four approaches: sliding windows of size 10 and 31 and averaging the 10 and 20 residues with the highest predicted propensities. We compare results generated using each of the 37 predictors and select the best method for each of the four protein-level scores, i.e., the method that secures the highest AUC and AUPRC values. We compare these best AUC scores (green bars) and best AUPRC scores (blue bars) between the four protein-level scores listed on the *x*-axis.

**Figure 3 biomolecules-14-00287-f003:**
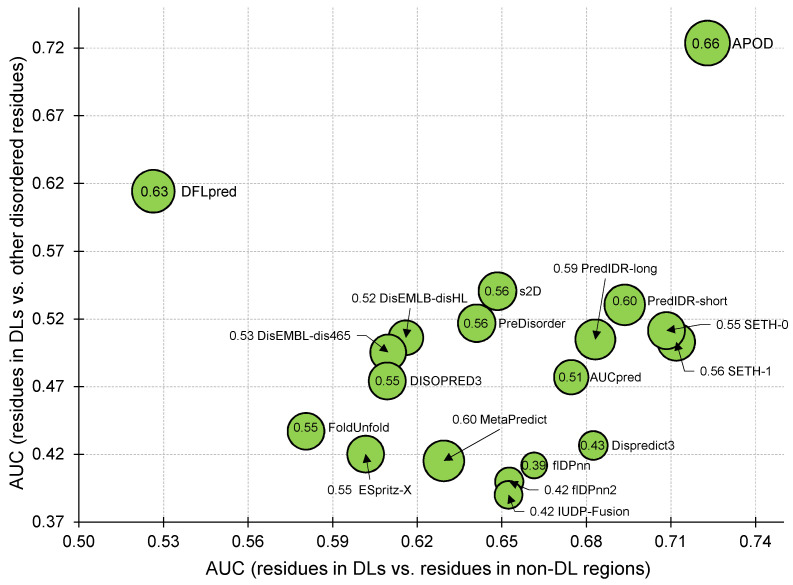
Predictive performance for the top ten methods that we identify using AUC scores in Table 1, Table 2 and Table 3. The x-axis shows the AUC when predicting residues in DLs vs. residues in non-DL regions (based on Table 1). The y-axis gives the AUC when predicting residues in DLs vs. other disordered residues (based on Table 2). The radius of the markers corresponds to the AUC for the predictions of proteins harboring DLs, which is listed inside the marker (based on Table 3). Methods are named next to their markers.

**Table 1 biomolecules-14-00287-t001:** Predictive performance for the prediction of residues in DLs on the CAID2 test dataset with 348 proteins. Predictors are sorted using their AUC values. We report averages from the ten test experiments. Performance metrics that are significantly different from the results of the most accurate method listed at the top of the table are denoted by + (*p*-value < 0.05); = means that the difference is not significant (*p*-value ≥ 0.05). We explain the metrics and test procedure in the *Materials and Methods* section.

Predictors	AUC	lowAUCratio	AUPRC	F1max	MCCmax
APOD	0.723	3.82	0.292	0.381	0.281
SETH-1	0.712 ^=^	2.70 ^+^	0.241 ^+^	0.349 ^+^	0.241 ^+^
SETH-0	0.708 ^=^	2.96 ^+^	0.257 ^+^	0.340 ^+^	0.230 ^+^
PredIDR-short	0.694 ^+^	3.15 ^+^	0.246 ^+^	0.341 ^+^	0.244 ^+^
PredIDR-long	0.683 ^+^	2.90 ^+^	0.233 ^+^	0.337 ^+^	0.246 ^+^
Dispredict3	0.682 ^=^	1.61 ^+^	0.205 ^+^	0.346 ^+^	0.234 ^+^
AUCpreD	0.675 ^+^	1.94 ^+^	0.210 ^+^	0.328 ^+^	0.207 ^+^
flDPnn	0.661 ^+^	1.81 ^+^	0.204 ^+^	0.340 ^+^	0.225 ^+^
flDPnn2	0.653 ^+^	1.91 ^+^	0.200 ^+^	0.330 ^+^	0.214 ^+^
IDP-Fusion	0.652 ^+^	1.51 ^+^	0.193 ^+^	0.328 ^+^	0.204 ^+^
s2D	0.648 ^+^	1.72 ^+^	0.189 ^+^	0.302 ^+^	0.171 ^+^
flDPlr2	0.646 ^+^	2.48 ^+^	0.215 ^+^	0.338 ^+^	0.221 ^+^
DeepIDP-2L	0.642 ^+^	1.56 ^+^	0.197 ^+^	0.340 ^+^	0.222 ^+^
PreDisorder	0.641 ^+^	3.00 ^+^	0.262 ^=^	0.350 ^+^	0.219 ^+^
RONN	0.640 ^+^	1.51 ^+^	0.185 ^+^	0.301 ^+^	0.171 ^+^
flDPtr	0.635 ^+^	1.58 ^+^	0.183 ^+^	0.301 ^+^	0.171 ^+^
IsUnstruct	0.631 ^+^	1.31 ^+^	0.178 ^+^	0.303 ^+^	0.178 ^+^
Metapredict	0.629 ^+^	0.52 ^+^	0.163 ^+^	0.296 ^+^	0.171 ^+^
DisoPred	0.621 ^+^	1.84 ^+^	0.192 ^+^	0.307 ^+^	0.179 ^+^
SPOT-Disorder-Single	0.620 ^+^	1.46 ^+^	0.176 ^+^	0.290 ^+^	0.151 ^+^
SPOT-Disorder	0.617 ^+^	0.91 ^+^	0.169 ^+^	0.301 ^+^	0.173 ^+^
DisEMBL-disHL	0.616 ^+^	1.75 ^+^	0.179 ^+^	0.278 ^+^	0.137 ^+^
DisoMine	0.615 ^+^	0.63 ^+^	0.161 ^+^	0.290 ^+^	0.159 ^+^
ESpritz-N	0.615 ^+^	1.63 ^+^	0.179 ^+^	0.280 ^+^	0.141 ^+^
MobiDB-lite	0.613 ^+^	1.74 ^+^	0.176 ^+^	0.282 ^+^	0.142 ^+^
DisEMBL-dis465	0.610 ^+^	1.76 ^+^	0.179 ^+^	0.279 ^+^	0.137 ^+^
DISOPRED3	0.609 ^+^	0.79 ^+^	0.163 ^+^	0.299 ^+^	0.164 ^+^
rawMSA	0.606 ^+^	1.70 ^+^	0.187 ^+^	0.323 ^+^	0.197 ^+^
VSL2	0.605 ^+^	0.67 ^+^	0.155 ^+^	0.290 ^+^	0.157 ^+^
IUPred3	0.602 ^+^	1.21 ^+^	0.165 ^+^	0.281 ^+^	0.143 ^+^
ESpritz-X	0.602 ^+^	1.25 ^+^	0.168 ^+^	0.276 ^+^	0.132 ^+^
AIUPred	0.595 ^+^	0.84 ^+^	0.160 ^+^	0.283 ^+^	0.145 ^+^
FoldUnfold	0.581 ^+^	1.39 ^+^	0.154 ^+^	0.263 ^+^	0.109 ^+^
Dispredict2	0.573 ^+^	1.22 ^+^	0.160 ^+^	0.274 ^+^	0.121 ^+^
pyHCA	0.569 ^+^	1.54 ^+^	0.165 ^+^	0.266 ^+^	0.136 ^+^
DFLpred	0.526 ^+^	1.51 ^+^	0.153 ^+^	0.235 ^+^	0.070 ^+^
ESpritz-D	0.512 ^+^	0.99 ^+^	0.138 ^+^	0.253 ^+^	0.109 ^+^

**Table 2 biomolecules-14-00287-t002:** Predictive performance for the prediction of residues in DLs vs. other disordered residues on the CAID2 test dataset with 348 proteins. Predictors are sorted using their AUC values. We report averages from the ten test experiments. Performance metrics that are significantly different from the results of the most accurate method listed at the top of the table are denoted by + (*p*-value < 0.05). We explain the metrics and test procedure in the *Materials and Methods* section.

Predictors	AUC	lowAUCratio	AUPRC	F1max	MCCmax
APOD	0.724	3.00	0.269	0.367	0.264
DFLpred	0.614 ^+^	1.63 ^+^	0.181 ^+^	0.279 ^+^	0.136 ^+^
s2D	0.541 ^+^	1.03 ^+^	0.142 ^+^	0.249 ^+^	0.076 ^+^
PredIDR-short	0.530 ^+^	2.17 ^+^	0.173 ^+^	0.290 ^+^	0.159 ^+^
PreDisorder	0.517 ^+^	1.98 ^+^	0.172 ^+^	0.270 ^+^	0.135 ^+^
SETH-0	0.512 ^+^	1.88 ^+^	0.167 ^+^	0.251 ^+^	0.108 ^+^
DisEMBL-disHL	0.506 ^+^	0.76 ^+^	0.128 ^+^	0.237 ^+^	0.043 ^+^
PredIDR-long	0.505 ^+^	2.11 ^+^	0.168 ^+^	0.284 ^+^	0.160 ^+^
SETH-1	0.503 ^+^	1.07 ^+^	0.136 ^+^	0.248 ^+^	0.077 ^+^
DisEMBL-dis465	0.495 ^+^	0.68 ^+^	0.126 ^+^	0.239 ^+^	0.046 ^+^
RONN	0.482 ^+^	0.61 ^+^	0.124 ^+^	0.241 ^+^	0.059 ^+^
AUCpreD	0.477 ^+^	1.01 ^+^	0.128 ^+^	0.236 ^+^	0.039 ^+^
DISOPRED3	0.474 ^+^	0.55 ^+^	0.120 ^+^	0.239 ^+^	0.052 ^+^
Dispredict2	0.465 ^+^	0.47 ^+^	0.114 ^+^	0.232 ^+^	0.055 ^+^
IsUnstruct	0.449 ^+^	0.46 ^+^	0.113 ^+^	0.236 ^+^	0.043 ^+^
ESpritz-N	0.447 ^+^	0.81 ^+^	0.119 ^+^	0.233 ^+^	0.048 ^+^
FoldUnfold	0.437 ^+^	0.83 ^+^	0.119 ^+^	0.186 ^+^	0.003 ^+^
DeepIDP-2L	0.427 ^+^	0.18 ^+^	0.109 ^+^	0.237 ^+^	0.042 ^+^
Dispredict3	0.427 ^+^	0.08 ^+^	0.106 ^+^	0.227 ^+^	0.042 ^+^
flDPlr2	0.421 ^+^	0.10 ^+^	0.108 ^+^	0.236 ^+^	0.051 ^+^
ESpritz-X	0.420 ^+^	0.80 ^+^	0.117 ^+^	0.234 ^+^	0.042 ^+^
MobiDB-lite	0.419 ^+^	0.30 ^+^	0.113 ^+^	0.206 ^+^	0.009 ^+^
Metapredict	0.415 ^+^	0.46 ^+^	0.112 ^+^	0.239 ^+^	0.063 ^+^
SPOT-Disorder-Single	0.414 ^+^	0.42 ^+^	0.109 ^+^	0.233 ^+^	0.040 ^+^
flDPnn	0.412 ^+^	0.01 ^+^	0.104 ^+^	0.235 ^+^	0.045 ^+^
VSL2	0.409 ^+^	0.19 ^+^	0.103 ^+^	0.234 ^+^	0.036 ^+^
rawMSA	0.401 ^+^	0.00 ^+^	0.103 ^+^	0.235 ^+^	0.035 ^+^
flDPnn2	0.400 ^+^	0.07 ^+^	0.102 ^+^	0.234 ^+^	0.043 ^+^
IUPred3	0.397 ^+^	0.62 ^+^	0.106 ^+^	0.233 ^+^	0.042 ^+^
IDP-Fusion	0.390 ^+^	0.11 ^+^	0.102 ^+^	0.236 ^+^	0.032 ^+^
SPOT-Disorder	0.383 ^+^	0.28 ^+^	0.101 ^+^	0.230 ^+^	0.030 ^+^
AIUPred	0.380 ^+^	0.50 ^+^	0.105 ^+^	0.233 ^+^	0.045 ^+^
pyHCA	0.368 ^+^	0.60 ^+^	0.105 ^+^	0.227 ^+^	0.028 ^+^
flDPtr	0.367 ^+^	0.10 ^+^	0.096 ^+^	0.231 ^+^	0.033 ^+^
DisoPred	0.360 ^+^	0.37 ^+^	0.101 ^+^	0.238 ^+^	0.031 ^+^
DisoMine	0.340 ^+^	0.04 ^+^	0.092 ^+^	0.233 ^+^	0.046 ^+^
ESpritz-D	0.282 ^+^	0.16 ^+^	0.087 ^+^	0.229 ^+^	0.033 ^+^

**Table 3 biomolecules-14-00287-t003:** Predictive performance for the prediction of proteins harboring DLs on the CAID2 test dataset with 348 proteins. We derive the protein-level scores using the maximum value of the sliding window-based average. The window size is 31 and corresponds to the median size of the DL regions. Predictors are sorted using their AUC values. We report averages from the ten test experiments. Performance metrics that are significantly different from the results of the most accurate method listed at the top of the table are denoted by + (*p*-value < 0.05); = means that the difference is not significant (*p*-value ≥ 0.05). We explain the metrics and test procedure in the *Materials and Methods* section.

Predictors	AUC	lowAUCratio	AUPRC	F1max	MCCmax
APOD	0.664	2.65	0.226	0.326	0.219
DFLpred	0.633 ^=^	1.75 ^+^	0.191 ^+^	0.280 ^+^	0.171 ^+^
Metapredict	0.605 ^+^	1.99 ^+^	0.179 ^+^	0.288 ^+^	0.173 ^+^
PredIDR-short	0.603 ^+^	1.09 ^+^	0.179 ^+^	0.304 ^=^	0.186 ^=^
PredIDR-long	0.594 ^+^	0.99 ^+^	0.173 ^+^	0.299 ^=^	0.179 ^=^
s2D	0.562 ^+^	1.66 ^=^	0.188 ^=^	0.263 ^+^	0.141 ^+^
PreDisorder	0.559 ^+^	2.41 ^=^	0.203 ^=^	0.246 ^+^	0.149 ^+^
SETH-0	0.558 ^+^	1.22 ^+^	0.164 ^+^	0.242 ^+^	0.112 ^+^
SETH-1	0.551 ^+^	0.51 ^+^	0.144 ^+^	0.247 ^+^	0.106 ^+^
DISOPRED3	0.550 ^+^	0.83 ^+^	0.138 ^+^	0.258 ^+^	0.125 ^+^
ESpritz-X	0.548 ^+^	1.25 ^+^	0.159 ^+^	0.253 ^+^	0.124 ^+^
FoldUnfold	0.547 ^+^	1.08 ^+^	0.133 ^+^	0.250 ^+^	0.136 ^+^
IsUnstruct	0.540 ^+^	1.31 ^+^	0.142 ^+^	0.249 ^+^	0.106 ^+^
DisEMBL-dis465	0.534 ^+^	0.45 ^+^	0.134 ^+^	0.251 ^+^	0.114 ^+^
rawMSA	0.533 ^+^	0.39 ^+^	0.134 ^+^	0.252 ^+^	0.111 ^+^
ESpritz-N	0.533 ^+^	0.54 ^+^	0.140 ^+^	0.243 ^+^	0.098 ^+^
SPOT-Disorder	0.524 ^+^	0.57 ^+^	0.141 ^+^	0.235 ^+^	0.091 ^+^
DisEMBL-disHL	0.518 ^+^	0.30 ^+^	0.130 ^+^	0.241 ^+^	0.088 ^+^
RONN	0.517 ^+^	0.40 ^+^	0.130 ^+^	0.242 ^+^	0.097 ^+^
AUCpred	0.514 ^+^	1.04 ^+^	0.126 ^+^	0.224 ^+^	0.050 ^+^
MobiDB-lite	0.513 ^+^	1.19 ^+^	0.137 ^+^	0.241 ^+^	0.106 ^+^
SPOT-Disorder-Single	0.503 ^+^	0.06 ^+^	0.123 ^+^	0.236 ^+^	0.076 ^+^
pyHCA	0.498 ^+^	1.04 ^+^	0.136 ^+^	0.233 ^+^	0.084 ^+^
flDPlr2	0.494 ^+^	0.08 ^+^	0.124 ^+^	0.236 ^+^	0.088 ^+^
VSL2	0.488 ^+^	0.09 ^+^	0.118 ^+^	0.237 ^+^	0.087 ^+^
AIUPred	0.481 ^+^	0.37 ^+^	0.120 ^+^	0.232 ^+^	0.077 ^+^
IUPred3	0.479 ^+^	0.10 ^+^	0.117 ^+^	0.231 ^+^	0.073 ^+^
DisoPred	0.470 ^+^	0.21 ^+^	0.119 ^+^	0.225 ^+^	0.048 ^+^
DeepIDP-2L	0.436 ^+^	0.00 ^+^	0.105 ^+^	0.229 ^+^	0.068 ^+^
DisoMine	0.435 ^+^	0.00 ^+^	0.109 ^+^	0.223 ^+^	0.047 ^+^
Dispredict3	0.430 ^+^	0.00 ^+^	0.105 ^+^	0.219 ^+^	0.037 ^+^
flDPnn2	0.424 ^+^	0.03 ^+^	0.106 ^+^	0.219 ^+^	0.035 ^+^
ESpritz-D	0.421 ^+^	0.24 ^+^	0.108 ^+^	0.214 ^+^	0.026 ^+^
IDP-Fusion	0.417 ^+^	0.34 ^+^	0.113 ^+^	0.228 ^+^	0.057 ^+^
flDPtr	0.405 ^+^	0.00 ^+^	0.101 ^+^	0.221 ^+^	0.041 ^+^
flDPnn	0.389 ^+^	0.00 ^+^	0.097 ^+^	0.219 ^+^	0.036 ^+^
Dispredict2	0.383 ^+^	0.04 ^+^	0.103 ^+^	0.221 ^+^	0.037 ^+^

## Data Availability

The underlying data is freely available at https://caid.idpcentral.org/challenge, accessed on 5 October 2023.

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
