# Peer review of "Assessment of Disordered Linker Predictions in the CAID2 Experiment"

_biomolecules, 2024, doi:10.3390/biom14030287_

Round 1
Reviewer 1 Report (Previous Reviewer 3)
Comments and Suggestions for Authors
The authors have addressed my concerns with the original manuscript.
However, there was another reviewer that also had some concerns originally, but I don't find a response to those? Were those concerns addressed too, or rebutted? That is my remaining concern, albeit a new one. I would like to know that the authors addressed the concerns of all of the reviewers.
Author Response
The authors have addressed my concerns with the original manuscript.
REPLY: Many thanks for your feedback.
However, there was another reviewer that also had some concerns originally, but I don't find a response to those? Were those concerns addressed too, or rebutted? That is my remaining concern, albeit a new one. I would like to know that the authors addressed the concerns of all of the reviewers.
REPLY: We addressed all concerns from the reviewers and provided the corresponding responses with the previous revision. Apparently, the publisher limits your ability to see all submitted responses. The bottom line is that we were asked to submit this subsequent revision that focuses solely on fixing language issues, and with no further concerns related to the original/previous review.
Reviewer 2 Report (New Reviewer)
Comments and Suggestions for Authors
The authors performed an assessment of disordered linker predictions based on the
CAID2 Experiment, where they expand the test set and investigated three scenarios: 1) prediction of residues in DLs vs. in non-DLs; 2) prediction of residues in DLs vs. in other disordered regions; 3) prediction of proteins harboring DLs. For the third scenario, the authors also created a practical metric to evaluate—at the protein level—the chance of a protein harboring DLs based on residue-level predictions. The material and methods look complete and the data representation is clear.
However, the writing is concerning and needs a lot of improvement in order to be accepted. There are two classes of issues in writing: 1) the choices of words are not accurate; 2) the choices of words are not consistent. I will point out several examples in each class; however, be sure to go through the whole manuscript and fix all the similar issues even if I haven’t pointed out.
Examples for the choices of words being inaccurate:
1. “Prediction of DLs vs. other residues”. I think you are not making the decision between a whole LD region vs other individual residue. You are actually trying to predict whether a given residue belongs to DLs or non-DL regions. Wouldn’t “prediction of residues in DLs vs. in non-DLs” be a better wording?
2. Line 79: “These methods were trained…” How can methods be trained? Do you want to say “these predictors were trained…” or “these models were trained…”?
3. Line 88: “deposited”? What does this mean? Do you actually want to say “assigned” or “distributed”?
4. Line 110-112: “Third, we assess the ability to identify proteins that have DLs from among other proteins that do not have disordered linkers.” Do you try to say “Third, we assess whether those tools are able to distinguish proteins harboring DLs from other proteins that do not harbor DLs”?
5. Line 159: “the number of the DL predictions”. Do you want to say “value” or “rate” rather than “number”?
6. Line 176: “we perform tests of statistical significance of differences between…”. People usually don’t say “statistical significance of differences”. Are you simply trying to say “we perform of statistical significance tests between…”?
7. Line 181: “to test normality of the resulting measurements”. What is “normality”??
8. Line 214 “Panels A and B concern prediction of DLs in sequences…” What does “concern” mean here?
9. Line 260-261: “DLs form sequence region…”. “sequence region” is not a common scientific term. Don’t you just want to say “DLs range between 10 and 288 consecutive residues in length”?
10. Line 273: “above-defined” is not common English.
Examples for the choices of words being inconsistent:
1. “predictors of DLs”, “predictors of disordered linkers”, “DL predictors”, “linker predictors”. There are at least four versions of these terms referring to the same concept, which make the readers very confused. Please stick to one, e.g., “DL predictors”.
2. “putative propensities”, “predicted real-valued propensities”, “predicted propensities”, “values of the propensities”, “propensity values”. Are all these terms actually the same or slightly different indeed? Please be clear and unify the expression.
3. “entropic disordered regions”, “entropic regions”, “entropic chain”. Are they referred to the same concept? Please use the most accurate term commonly accepted in the field and unify the expression.
4. In line 125: “We consider all methods that participated in CAID2. However, we eliminate tools…” What’s the difference between methods and tools? Are they the same concept indeed? Please unify the expression.
Other issues:
1. In line 115-117: “This means that disorder predictors might be able to predict DLs with higher putative propensities for disorder than other types of IDRs that undergo some form of folding”. In line 198-200: “The low predictive performance of many of these tools is not surprising since they were designed to predict disordered residues rather than DLs.” Are these two sentences contradictory to some degree?
2. In line 32-34: “While cellular functions of many IDRs involve transition into structured state(s), the entropic disordered regions function by shifting between conformational states without becoming structured”. Need a reference here.
3. In line 52-53: “Although it may seem that any IDR can act as a linker [17], it is clearly a mistake to assume that all IDRs are made equal.” I don’t think people are assuming that all IDRs are made equal at all. Please rewrite the first sentence of this paragraph.
4. Line 128-131. I couldn’t understand. Please rewrite the sentence.
5. Line 171-176: couldn’t understand. Please rewrite the sentence.
6. Figure 3 x-axis: “linker vs non-linker”. Shouldn’t it be “DLs vs non-DLs”? Are you testing linkers in general here or are only testing disordered linkers? Same issue for the y-axis.
7. I suggest using “prediction of proteins harboring DLs” to replace “prediction of proteins with DLs” in the whole manuscript.
8. The authors proposed a third scenario of “prediction of proteins harboring DLs”. This is an interesting task, but why is this important? To be specific, I can imagine knowing which part of the protein being DL at residue level is important for me to understand the protein function, but how can having the binary information of whether a protein harboring DLs at protein level help me understand better about the protein function? Please add extra sentences in appropriate position in discussion to address this question.

See attachment
Author Response
The authors performed an assessment of disordered linker predictions based on the CAID2 Experiment, where they expand the test set and investigated three scenarios: 1) prediction of residues in DLs vs. in non-DLs; 2) prediction of residues in DLs vs. in other disordered regions; 3) prediction of proteins harboring DLs. For the third scenario, the authors also created a practical metric to evaluate—at the protein level—the chance of a protein harboring DLs based on residue-level predictions. The material and methods look complete and the data representation is clear.
REPLY: Many thanks for your feedback.
However, the writing is concerning and needs a lot of improvement in order to be accepted. There are two classes of issues in writing: 1) the choices of words are not accurate; 2) the choices of words are not consistent. I will point out several examples in each class; however, be sure to go through the whole manuscript and fix all the similar issues even if I haven’t pointed out.
REPLY: Thanks for your detailed suggestions. We implemented them to the best of our ability.
Examples for the choices of words being inaccurate:
1. Prediction of DLs vs. other residues”. I think you are not making the decision between a whole LD region vs other individual residue. You are actually trying to predict whether a given residue belongs to DLs or non-DL regions. Wouldn’t “prediction of residues in DLs vs. in non-DLs” be a better wording?
REPLY: Fixed per your suggestion.
2. Line 79: “These methods were trained…” How can methods be trained? Do you want to say “these predictors were trained…” or “these models were trained…”?
REPLY: Fixed per your suggestion.
3. Line 88: “deposited”? What does this mean? Do you actually want to say “assigned” or “distributed”?
REPLY: They were deposited (code was given to the assessors who run the code) but we have changed it to “provided”
4. Line 110-112: “Third, we assess the ability to identify proteins that have DLs from among other proteins that do not have disordered linkers.” Do you try to say “Third, we assess whether those tools are able to distinguish proteins harboring DLs from other proteins that do not harbor DLs”?
REPLY: Fixed per your suggestion.
5. Line 159: “the number of the DL predictions”. Do you want to say “value” or “rate” rather than “number”?
REPLY: We clarified it by writing: “the number of residues predicted to be in DLs is relatively low (conservative) and does not exceed the number of the native DL residues.”
6. Line 176: “we perform tests of statistical significance of differences between…”. People usually don’t say “statistical significance of differences”. Are you simply trying to say “we perform of statistical significance tests between…”?
REPLY: Fixed per your suggestion (excluding “of”) although the original explanation was more precise.
7. Line 181: “to test normality of the resulting measurements”. What is “normality”??
REPLY: Fixed as “to test whether resulting measurements are normal”.
8. Line 214 “Panels A and B concern prediction of DLs in sequences…” What does “concern” mean here?
REPLY: Fixed as “Panels A and B focus on the assessment of the predictions of DL residues”
9. Line 260-261: “DLs form sequence region…”. “sequence region” is not a common scientific term. Don’t you just want to say “DLs range between 10 and 288 consecutive residues in length”?
REPLY: Fixed per your suggestion.
10. Line 273: “above-defined” is not common English.
REPLY: Fixed as “that we define above”
Examples for the choices of words being inconsistent:
1. “predictors of DLs”, “predictors of disordered linkers”, “DL predictors”, “linker predictors”. There are at least four versions of these terms referring to the same concept, which make the readers very confused. Please stick to one, e.g., “DL predictors”.
REPLY: Fixed per your suggestion.
2. “putative propensities”, “predicted real-valued propensities”, “predicted propensities”, “values of the propensities”, “propensity values”. Are all these terms actually the same or slightly different indeed? Please be clear and unify the expression.
REPLY: Fixed per your suggestion.
3. “entropic disordered regions”, “entropic regions”, “entropic chain”. Are they referred to the same concept? Please use the most accurate term commonly accepted in the field and unify the expression.
REPLY: Fixed as “entropic disordered regions”
4. In line 125: “We consider all methods that participated in CAID2. However, we eliminate tools…” What’s the difference between methods and tools? Are they the same concept indeed? Please unify the expression.
REPLY: Unified to “methods”
Other issues:
1. In line 115-117: “This means that disorder predictors might be able to predict DLs with higher putative propensities for disorder than other types of IDRs that undergo some form of folding”. In line 198-200: “The low predictive performance of many of these tools is not surprising since they were designed to predict disordered residues rather than DLs.” Are these two sentences contradictory to some degree?
REPLY: The first sentence poses a hypothesis by stating “might be able”. The second sentence is an observation based on results. We rephrased the second sentence to minimize the potential notion of a contradiction as follows: “The low predictive performance of many of these methods is due to the fact that they were designed to predict disordered residues and apparently their predictions cannot be used to accurately predict DL residues from among other disordered and structured residues.”
2. In line 32-34: “While cellular functions of many IDRs involve transition into structured state(s), the entropic disordered regions function by shifting between conformational states without becoming structured”. Need a reference here.
REPLY: We added the references.
3. In line 52-53: “Although it may seem that any IDR can act as a linker [17], it is clearly a mistake to assume that all IDRs are made equal.” I don’t think people are assuming that all IDRs are made equal at all. Please rewrite the first sentence of this paragraph.
REPLY: This was in response to a reviewer that implied exactly that. However, we fixed the sentence to say: “Although it may seem that any IDR can act as a linker [17], the levels and depth of disorder in proteins can be very different, with different “disorder flavors” being dis-tinguished based on the differences in amino acid compositions, sequence locations, and biological functions [18].”
4. Line 128-131. I couldn’t understand. Please rewrite the sentence.
REPLY: We rewrote and simplified the sentence to make it clearer.
5. Line 171-176: couldn’t understand. Please rewrite the sentence.
REPLY: We rewrote and simplified these sentences to make them clearer.
6. Figure 3 x-axis: “linker vs non-linker”. Shouldn’t it be “DLs vs non-DLs”? Are you testing linkers in general here or are only testing disordered linkers? Same issue for the y-axis.
REPLY: We fixed descriptions for both x-axis and y-axis.
7. I suggest using “prediction of proteins harboring DLs” to replace “prediction of proteins with DLs” in the whole manuscript.
REPLY: Fixed per your suggestion.
8. The authors proposed a third scenario of “prediction of proteins harboring DLs”. This is an interesting task, but why is this important? To be specific, I can imagine knowing which part of the protein being DL at residue level is important for me to understand the protein function, but how can having the binary information of whether a protein harboring DLs at protein level help me understand better about the protein function? Please add extra sentences in appropriate position in discussion to address this question.
REPLY: We motivate this scenario in the latter part of the Introduction section where we introduce it. We say: “This is a useful scenario for methods that struggle with predicting correct positions of DL residues but which accurately identify presence of these residues in a given sequence, especially if they make these predictions very quickly, facilitating applications to large collections of proteins.” We also briefly discuss this aspect in the “Summary and Conclusions” section where we say: “Accurate predictors of proteins harboring DLs would be particularly useful if their results could be produced very quickly, allowing for fast screening of proteins for presence of DLs. Such fast predictions could be followed up by running a relatively slow APOD, which takes about 560 seconds to predict 1000 residues long sequence [33] and which accurately identifies residues in DLs.”
Round 2
Reviewer 2 Report (New Reviewer)
Comments and Suggestions for Authors
Good improvement in writing.
This manuscript is a resubmission of an earlier submission. The following is a list of the peer review reports and author responses from that submission.
Round 1
Reviewer 1 Report
Comments and Suggestions for Authors
The study is interesting and well written.
The study evaluates predictions of disordered linkers (DLs) using various predictive models, incorporating assessment criteria not present in CAID2. The study reveals that the APOD tool's predictions for DLs outperformed those of other models, highlighting the need for additional tools that can achieve higher accuracy in predicting DLs.
Reviewer 2 Report
Comments and Suggestions for Authors
Wang et al. aims to benchmark the computational prediction of disordered linkers using a new dataset derived from a community exercise in evaluation of disorder predictors. They mostly benchmark their own predictors, APOD and DFLpred, against general predictors of disorder, of which the best is only marginally better at predicting disordered linkers than APOD despite not explicitly trained for this purpose. The performance of the best predictor is modest at best with only a little more than two-fold better performance than random. No new tools are developed here, only a questionable benchmarking of already existing tools.
The premise of the study seems to be based on some rather weird assumptions, that disordered linkers are a distinct subset of IDRs with identifiable sequence features that can be used to train a ML model. This is in contradiction with basically all the experimental data in the field with suggests that basically any IDR can act a linker. See for example reference 7, where hundreds of synthetic sequences are shown to work as linkers. Evolution of linkers is dependent on the context in which they evolve and compaction and length can compensate for each other (e.g. https://doi.org/10.1038/s41594-022-00811-w)
The annotation of IDRs used in the training sets is highly variable and is often a negative category – i.e. that IDRs get annotated as “linker” when there are no discernable functions. It is unclear what the predictors actually predict, and the authors description does not help much as they do not really describe what they consider a linker except than uncritically accepting the annotations in databases. Also, several times they refer to the annotations as the “ground truth”. This suggests a rather worrying nativity about the quality of the data used both in the evaluation and in the training of the predictors.
Reviewer 3 Report
Comments and Suggestions for Authors
I have 2 main concerns:
1. Background information on DLs (disordered linkers) is sparse in the introduction. The authors gave a much more complete discription of DLs in Bioinformatics, 36(26), 2020, i754–i761. The introduction to the current manuscript should strive to give a similar level of detail as that provided in their 2020 article.
2. The idea of the manuscript is to test various predictors in ability to successfully identify DLs within protein sequences. It seems disingenuous to include predictors that were not designed to identify DLs (i.e., only 2 of the 41 evaluated predictors were designed to identify DLs). As written, the manuscript does a poor job of justifying the use of disorder-only predictors in their study. This needs to be fixed to distinguish this article from the numerous other recent and published studies that have tested ID predictors.
Comments on the Quality of English LanguageMostly fine.